# Endo-β-1,3-glucanase (GH16 Family) from *Trichoderma harzianum* Participates in Cell Wall Biogenesis but Is Not Essential for Antagonism Against Plant Pathogens

**DOI:** 10.3390/biom9120781

**Published:** 2019-11-26

**Authors:** Marcela Suriani Ribeiro, Renato Graciano de Paula, Aline Raquel Voltan, Raphaela Georg de Castro, Cláudia Batista Carraro, Leandro José de Assis, Andrei Stecca Steindorff, Gustavo Henrique Goldman, Roberto Nascimento Silva, Cirano José Ulhoa, Valdirene Neves Monteiro

**Affiliations:** 1Department of Biochemistry and Cellular Biology, Biological Sciences Institute, Campus Samambaia, Federal University of Goiás (UFG), Goiânia, Goiás 74090-900, Brazil; marcelasuriani@gmail.com (M.S.R.); alinervoltan@gmail.com (A.R.V.); rcgeorg@gmail.com (R.G.d.C.); 2Department of Physiological Sciences, Health Sciences Centre, Federal University of Espirito Santo, Vitoria ES 29047-105, Brazil; renato.gracciano@gmail.com; 3Department of Biochemistry and Immunology, Ribeirão Preto Medical School, University of São Paulo (USP), Ribeirão Preto 14049-900, Brazil; carraro.clau@gmail.com; 4Departamento de Ciências Farmacêuticas, Faculdade de Ciências Farmacêuticas de Ribeirão Preto, Universidade de São Paulo, Av. do Café S/N, Ribeirão Preto CEP 14040-903, Brazil; ljassis@icloud.com (L.J.d.A.); ggoldman@usp.br (G.H.G.); 5Department of Cell Biology, University of Brasília, Campus Darcy Ribeiro, Biological Sciences Institute, Brasília CEP 70.910-900, Brazil; andreistecca@gmail.com; 6Campus of Exact Sciences and Technologies, Campus Henrique Santillo, State University of Goiás (UEG), BR 153, Km 98, Anápolis 75000-000, Brazil

**Keywords:** *Trichoderma harzianum*, endo-β-1,3-glucanase, cell wall, glycosyl hydrolase family 16, *gluc31* gene

## Abstract

*Trichoderma* species are known for their ability to produce lytic enzymes, such as exoglucanases, endoglucanases, chitinases, and proteases, which play important roles in cell wall degradation of phytopathogens. β-glucanases play crucial roles in the morphogenetic-morphological process during the development and differentiation processes in *Trichoderma* species, which have β-glucans as the primary components of their cell walls. Despite the importance of glucanases in the mycoparasitism of *Trichoderma* spp., only a few functional analysis studies have been conducted on glucanases. In the present study, we used a functional genomics approach to investigate the functional role of the *gluc31* gene, which encodes an endo-β-1,3-glucanase belonging to the GH16 family in *Trichoderma harzianum* ALL42. We demonstrated that the absence of the *gluc31* gene did not affect the in vivo mycoparasitism ability of mutant *T. harzianum* ALL42; however, *gluc31* evidently influenced cell wall organization. Polymer measurements and fluorescence microscopy analyses indicated that the lack of the *gluc31* gene induced a compensatory response by increasing the production of chitin and glucan polymers on the cell walls of the mutant hyphae. The mutant strain became more resistant to the fungicide benomyl compared to the parental strain. Furthermore, qRT-PCR analysis showed that the absence of *gluc31* in *T. harzianum* resulted in the differential expression of other glycosyl hydrolases belonging to the GH16 family, because of functional redundancy among the glucanases.

## 1. Introduction

The genus *Trichoderma* comprises species that exhibit agonistic activity against plant pathogens, are involved in plant disease prevention, and can influence plant growth and development [1,2]. *Trichoderma* species are known for their ability to produce lytic enzymes, including exoglucanases, endoglucanases, chitinases, and proteases, which play important roles in cell wall degradation of phytopathogens [3,4,5].

The fungal cell wall is generally composed of 90% polysaccharides (glucan and chitin), which are the main polymers of carbohydrates, as well as other polymers, such as α-1,3-glucan, α-1,4-glucan, β-1,3-glucan, β-1,3-1,4-glucan, β-1,6-glucan, chitosan, mannan, and galactomannan. The fungal cell wall components vary depending on the stage of growth, species, and morphotypes [1,6,7].

β-glucanases play crucial roles in the morphogenesis process during development and differentiation in *Trichoderma* species, which have β-glucans as the primary component of their cell walls [8]. Glucanases can hydrolyze their substrates via the following two possible mechanisms: (i) exo-β-1,3-glucanases (EC3.2.1.58) that hydrolyze β-glucans by sequentially cleaving glucose residues from the non-reducing end and (ii) endo-β-1,3-glucanases (EC3.2.1.39) that cleave β-linkages at random sites along the polysaccharide chain, leading to the release of smaller oligosaccharides and glucose [9].

Sequencing and annotation of different *Trichoderma* genomes have enabled the functional analysis of the genes, such as selection of candidate genes and construction of DNA vectors and deletion cassettes [10,11].

Different genes identified in *Trichoderma* spp. have been deleted, silenced, or overexpressed in previous studies [12]. These include cell signaling genes [13,14], plant resistance elicitors [15], and genes encoding hydrolytic enzymes, such as chitinases and proteases [16,17]. Despite the importance of glucanases in the mycoparasitism of *Trichoderma* species, only a few studies have conducted a functional analysis of glucanases. Previous studies have reported the deletion and overexpression of a β-1,6-glucanase (tvbgn3) from *Trichoderma virens* and overexpression of a β-1,4-glucanase encoded by the *egl1* gene from *Trichoderma longibrachiatum* [18,19]. Transcriptome studies of the interactions between *T. harzianum* and *Fusarium solani* have investigated genes encoding cell wall degradation enzymes that are involved in mycoparasitism. Of these, the *bgn13.4* gene, which encodes a β-1,3-glucanase, was found to be upregulated during the later stages of the interaction between *Trichoderma* and the phytopathogen [20,21]. Thus, it is of great interest to understand the physiological relevance of glucanases for the *T. harzianum* ALL42 species, especially to elucidate its role in biocontrol.

Furthermore, current knowledge about β-1,3-glucanases remain limited to biochemical studies and determination of the properties of the purified enzymes. Therefore, we performed a functional genomics approach to study the functional role of the *gluc31* gene, which encodes an endo-β-1,3-glucanase of the GH16 family in *Trichoderma harzianum* ALL42.

## 2. Materials and Methods

### 2.1. Culture Conditions

*Trichoderma harzianum* (ALL42) was obtained from the collection of the Laboratory of Enzymology of Universidade Federal de Goiás, Brazil. The ITS1-5.8S-ITS2 region and 5th intron of *tef1* were sequenced (HQ857122 and HQ857136, respectively) to confirm *T. harzianum* identification. The phytopathogenic fungal strains *Sclerotinia sclerotiorum*, *Rhizoctonia solani*, and *Fusarium oxysporum* were provided by Embrapa (The Brazilian Agricultural Research Corporation, Rice and Bean unit, Santo Antônio de Goiás, Brazil), maintained with periodical sampling on potato dextrose agar (potato dextrose-agar medium (Difco, Detroit, MI, USA), and stored at 4 °C. The ITS regions of the wild-type *T. harzianum* ALL42 and Δ*gluc31* mutant were sequenced for species confirmation.

### 2.2. Selection of the Gene for Knockout Studies and In Silico Analysis

The selected gene encodes the endo-β-1,3-glucanase protein (ID 150678) of *T. harzianum*. The amino acid and nucleotide sequence of the genes analyzed in this study were obtained from the *T. harzianum* database (http://genome.jgi.doe.gov/Triha1). The Pfam v29.0 database was used to identify the protein family. The InterPro Hidden Markov models (HMM) program was used for sequence analysis. The servers can be found on the European Bioinformatics Institute (EMBL-EBI, Hinxton, Cambridgeshire, United Kingdom) (https://www.ebi.ac.uk/services). The programs SignalP 4.1, NetNGlic 1.0, YinOYang 1.2, and (Trans-membrane Hidden Markov models) TMHMM 2.0 (SIB Swiss Institute of Bioinformatics, Lausanne, Switzerland) were used for the prediction of cell secretion signals, *N*-glycosylation signals, *O*-glycosylation, and transmembrane helices, respectively. The above-mentioned tools can be found in the Center for Biological Sequence Analysis (http://www.cbs.dtu.dk/biotools/).

### 2.3. Construction of the Δgluc31 T. harzianum Mutant Strain

The construction of the mutant strain was performed according to the protocol described in [22]. The deletion cassette was constructed via fusion PCR using the primers listed in Appendix A by using a selection hph cassette (*hph* gene, which confers resistance to hygromycin and is under the control of the *Aspergillus nidulans* gpdA promoter and trpC terminator) according to the method described by Mach et al. (1994) [23]. Fungal transformations were performed using protoplasts according to the methods of [24]. Transformants were selected via three rounds of homokaryon isolation and resistance to hygromycin B (100 µg/mL).

### 2.4. Characterization of Transformants

The selected transformants were analyzed via Southern hybridization as previously described [25] to verify homologous transformation of the cassettes into the target *T. harzianum* loci. Briefly, 25 µg of total genomic DNA from both parental and mutant strains were digested overnight with *Eco*RI and *Xba*I (Fermentas). Afterwards, the, digested DNA was transferred to Amersham Hybond-N+ membranes (GE healthcare, Chicago, IL, USA). The probe was produced from the PCR-amplified fragment using Del_5F/hph_R primers (Appendix A) and subsequently labeled using a digoxigenin (DIG) DNA labeling kit (Roche, Mannheim, Germany) following the manufacturer’s instructions. Labeling, hybridization, and immunological detection were carried out using a non-radioactive labeling and immunological detection kit and CDP-Star as the chemiluminescent substrate (Roche, Mannheim, Germany) as previously described [25]. Moreover, homologous integration of the disruption cassette was evaluated via PCR using primers specific to the *hph* gene in combination with primers specific to the sequences flanking the deletion construct (Del_5F/Hph_3R). Finally, a gene expression analysis of the *T. harzianum* wild-type (WT) and *T. harzianum Δgluc31* strains was performed via qRT-PCR using primers specific to *gluc31* and β-actin genes (Appendix A).

### 2.5. Gene Expression Analysis

The mycelia collected from three independent experiments for each condition were used for qRT-PCR analyses. For the first experiment, the interactions between *T. harzianum* wild-type and the pathogen *R. solani* were analyzed. *F. oxysporum* and *S. sclerotiorum* were grown in PDA agar plates covered with cellophane at 28 °C under contact and post-contact conditions (described in Section 2.8), after which *gluc31* expression was evaluated. For the second experiment, culture discs of *T*. *harzianum* WT and *T. harzianum Δgluc31* strains were grown on PDA agar plates covered with cellophane at 28 °C, and the mycelia were collected after 7 days for expression analysis of the chitin synthase and glucan synthase genes. Finally, the third experiment evaluated the interactions between wild-type *T*. *harzianum* and the phytopathogen *R. solani* and between *T. harzianum Δgluc31* and the phytopathogen *R. solani*. Thus, culture disks from each of the *T. harzianum* (wild and mutant) strains were added diametrically opposite to *R. solani* on PDA agar plates covered with cellophane at 28 °C under contact and post-contact conditions. The wild-type and mutant (*Δgluc31*) *T. harzianum* strains were also grown in PDA medium without the presence of phytopathogens at 28 °C for 7 days. Afterwards, mycelia were collected for the expression analysis of genes belonging to the GH16 family of β-1,3-glucanases from *T. harzianum* (Appendix A), which was identified based on genome annotation of *T. harzianum* available on the website http://genome.jgi.doe.gov/pages/search-for-genes.jsf?organism=Triha1. The mycelia collected from the three treatment groups were frozen in liquid nitrogen and macerated. Total RNA was extracted using the TRI-Reagent^®^ solution (Sigma-Aldrich, St. Louis, MI, USA) according to the manufacturer’s instructions.

For each sample, total RNA was digested with DNase I (Invitrogen, Carlsbad, CA, USA). A total of 5 μg of RNA from each of the pooled samples was reverse transcribed into cDNA using the Maxima™ (Thermo Fisher Scientific, Waltham, MA, USA) First Strand cDNA Synthesis Kit for qRT-PCR with a final volume of 20 μL (Thermo Fisher Scientific). The synthesized cDNA was diluted with 80 μL of water and used as a template for real-time PCR on the StepOnePlus^TM^ Real-Time PCR System (Thermo Fisher Scientific). Each reaction (20 μL) contained 10 μL of MAXIMA^®^ SYBR Green PCR Master Mix (Thermo Fisher Scientific), forward and reverse primers (500 nM each), a cDNA template, and nuclease-free water. PCR efficiency was determined based on triplicate reactions using a cDNA dilution series (100, 10^−1^, 10^−2^, and 10^−3^ dilutions). Amplification efficiency was then calculated based on the slopes given in the StepOnePlus^TM^ Software v2.3 (Thermo Fisher Scientific). β-actin transcript levels were used as an internal reference for normalization [20]. Gene expression levels were calculated from the threshold cycle following the 2-^ΔΔCT^ method [26]. All samples were analyzed in triplicate for each run.

### 2.6. Extraction of Cell Wall Polysaccharides and Sugar Quantification

Polysaccharide extraction was performed as previously described in [27] by using 10 mg of dry cell wall mass. After extraction, 1 mL of the final preparation was concentrated (10×) by lyophilization and used for sugar quantification via HPLC (Young Lin System, Anyang, Korea). Sugar levels were measured using a Young Lin YL9100 series system (Young Lin system, Anyang, Korea) equipped with a YL9170 series refractive index (RI) detector at 40 °C. The samples were loaded onto the REZEX ROA (Phenomenex, Torrance, CA, USA) column (300 × 7.8 mm) at 85 °C and eluted with 0.05 M sulfuric acid at a flow rate of 1.5 mL/min. Chitin levels were estimated based on *N*-acetyl-glucosamine. Glucan content was determined using the dectin-1 stain as described in [28] with minor modifications. Conidia (2 × 10^5^) from *T. harzianum* were grown for 36 h at 30 °C in MEX medium and the medium was removed. The mycelia were fixed using a solution containing 5% DMSO and 10% formaldehyde in 1X PBS buffer for 5 min at room temperature; washed once with PBS buffer; blocked using 200 µL of a blocking solution (goat serum 2%, BSA 1%, 0.1% Triton X-100, 0.05% Tween 20, 0.05% sodium azide, and 0.01 M PBS) for 30 min at room temperature; stained with PBS buffer containing 0.2 µg/mL of s-dectin-hFc-1a (Invitrogen, Carlsbad, CA, USA) for 1 h at room temperature; washed once with PBS buffer and followed by DyLight 594-conjugated (Abcam, Cambridge, UK) (dilution 1:1000) for 1 h; washed again with PBS buffer; and fluorescence was detected according to the manufacturer’s instructions at 587 nm and 615 nm wavelengths for excitation and emission, respectively.

### 2.7. Fluorescence Microscopy

Microcultures derived from the wild-type and mutant lineage culture dishes were conducted based on the modified Riddell technique (1950) [29]. After growing the cultures, the coverslips were removed from the culture dish and stained with 0.01% calcofluor white (Sigma-Aldrich) and examined under a Zeiss Imager M2 fluorescence microscope (Zeiss, Oberkochen, Germany) coupled to the AxioVision (Zeiss, Oberkochen, Germany) software. Emission was recorded under blue light excitation at 300–400 nm. After the images were acquired, they were submitted to an analysis of the wall thickness and width of the hyphae through the intensity of fluorescence emitted by the fungi. The analyses were performed using the Image-Pro Plus 6.0 program (Media Cybernetics, Inc., Rockville, MD, USA).

### 2.8. Direct Confrontation Assays

We evaluated the antagonism of the mutant and wild-type *Trichoderma* strains against the phytopathogens *S. sclerotiorum*, *R. solani*, and *F. oxysporum*. Discs containing PDA medium were obtained from the edges of actively growing colonies of fresh fungal cultures and subsequently placed on the surface of a fresh PDA plate at 8 cm apart. Plates were incubated at 28 °C with a photoperiod of 12 h. Antagonism was evaluated according to the classification proposed in [30].

### 2.9. Phylogeny of the Glycoside Hydrolase Family 16 in the Trichoderma Genus

First, all proteins containing the glycol_hydro_16 (PF00722) domain from *Trichoderma* genomes were collected from the MycoCosm portal [31], which contains a total of 100 proteins. A local version of InterProScan5 (https://www.ebi.ac.uk/interpro/interproscan.html) was used to predict the Pfam domains and signal peptides (SignalP4.1). The glycol_hydro_16 region of the protein was trimmed prior to alignment using Muscle [32] and a maximum likelihood phylogeny was constructed using fast tree [33]. The position of each gene feature was used as an input for Evolview [34] to visualize gene architecture.

### 2.10. Enzyme Activities

The enzymatic activities of chitinase, β-1,3-glucanase, and lichenase [35] were determined via the colorimetric method using colloidal chitin and laminarin as substrates. The amounts of reducing sugar released were determined at 550 nm following the DNS method using *N*-acetylglucosamine and glucose as standards. One unit of enzymatic activity (U) was defined as the amount of enzyme that released 1 µmol of reducing sugars per minute [36]. *N*-acetyl glycosaminidase activity was determined via the colorimetric method using PNP-NAG (Sigma-Aldrich) as the substrate [36]. Enzyme activity was measured by monitoring the rate of formation of *p*-nitrophenol from the substrate. One unit (U) of enzymatic activity was defined as the amount of enzyme required to release 1 µmol of *p*-nitrophenol per minute.

### 2.11. Alamar Blue Viability Assay

In order to test the resistance and viability of the fungal cells under treatment with benomyl (Sigma-Aldrich Co., Wisconsin, USA), 5 × 10^3^ conidia mL^−1^ from both ALL42 and Δ*gluc31* strains were inoculated into 200 μL of liquid potato dextrose medium (10% potato in distilled water) containing 10% alamarBlue^TM^ (Thermo Fischer Scientific) in a 96-well microplate. Triplicates for each benomyl concentration (0.0, 0.1, 0.25, 0.4, 0.5, 1.0, and 1.5 μg/mL) were used for each strain. The plate was incubated at 30 °C, and results were read on spectrophotometer in 570–600 nm after 24 h of incubation; and then, graphs containing the optical density for each test were plotted by using GraphPad Prism 7.04. Furthermore, we also performed an experiment to analyze the growth of *T. harzianum* parental (ALL42) and *Δgluc31* strains in PDA medium plates with benomyl as the selective agent.

### 2.12. Statistical Analysis

All experiments were conducted in biological triplicates and statistical analyses were performed with one-way ANOVA (non-parametric) followed by the Bonferroni test (comparing all pairs of columns) available in the GraphPad Prism 7.04 (GraphPad Software Inc, San Diego, CA, USA).

## 3. Results

### 3.1. Sequence Analysis of the gluc31 Gene

The *gluc31* gene sequence, which encodes a *T. harzianum* endoglucanase (ID 150678), was obtained from http://genome.jgi.doe.gov/pages/search-for-genes.jsf?organism=Triha1, in which the *T. harzianum* genome sequence is deposited (Appendix A). The *gluc31* gene has a length of 939 bp, with only one intron of length 60 bp. The *gluc31* gene was selected based on the alignment (Blastn) generated from the EST sequences (GenBank HS573580) reported in the work of Steindorff et al. (2012) [20]. An analysis using ProtParam (ExPASy) v30.0 showed that the Gluc31 protein had 293 amino acids, a molecular weight of 31505.68 Da, and a theoretical PI of 4.98. Gluc31 contained a cell secretion signal (SignalP 4.0 program) and conserved domains of the GH16 family (Pfam00722). In addition, Gluc31 exhibited homology to the superfamily concanavalin lectin A-like (InterPro HMM program) and contained a total of 13 putative *O*-β-glycosylation sites (Appendix A). No predicted *N*-glycosylation site was found (NetNGlic1.0) (Appendix A).

### 3.2. Deletion of the gluc31 Gene from T. harzianum

A linear fragment containing the hygromycin gene as a selection marker was constructed via fusion PCR to replace the ORF of the *gluc31* gene via homologous recombination (Appendix A). Stable transformants were evaluated via standard PCR to verify correct integration of the deletion fragment. The 2.68 kb band indicates amplification of the 5ʹ flanking region and hygromycin gene (Appendix A). Of the two transformants positive for the corresponding band, only one transformant showed successful deletion based on qRT-PCR and Southern hybridization (Figure 1C,D). A phenotypic analysis on the PDA medium showed that the *Δglu31* mutant exhibited less sporulation than the parental ALL42 strain (Figure 4A).

### 3.3. Gluc31 Is Involved in Cell Wall Remodeling of T. harzianum

The evaluation of the effects of Gluc31 on the cell wall of *T. harzianum* was performed by determining the glucan and chitin contents. The comparison of glucan and chitin contents between the mutant (Δ*glu31*) and wild-type (ALL42) strains revealed that the mutant strain (*Δglu31*) had a higher *N*-acetyl-glucosamine and β-1,3-glucan content (Figure 2F,H). Fluorescence microscopy analyses indicated the involvement of Gluc31 in the cell wall dynamics of *T. harzianum*. The hyphae impregnated with calcofluor demonstrated high chitin accumulation in the apical hyphae (Figure 2D). Furthermore, the absence of Gluc31 decreased the hyphal width (Figure 3A,C) and increased cell wall thickness (Figure 3B,D). The results suggested the involvement of Gluc31 in cell wall remodeling as a compensatory effect that promotes chitin and glucan accumulation in response to *gluc31* gene deletion.

### 3.4. Cell Viability Assay

Since chitin and glucan content was affected by *gluc31* gene deletion, we tested the antifungal resistance of *T. harzianum* parental ALL42 and *Δgluc31* strains under benomyl influence (Figure 4). The PDA plate assay showed that the *Δgluc3*1 mutant strain had an intrinsic ability to grow better in the presence of 0.5 µg/mL of benomyl when compared with the ALL42 parental strain (Figure 4A). This effect was evaluated during six days of growth and our results showed that from day 4 of growth, Δ*gluc31* mutant strain had a pronounced resistance to benomyl compared to the ALL42 parental strain (Figure 4B). Moreover, the experiment with Alamar blue was performed to show the effects of benomyl on the viability of *T. harzianum* parental ALL42 and *Δgluc31* strains. The *Δgluc31* strain was more resistant than the parental strain ALL42 in the liquid medium with benomyl (Figure 4C), similarly to what was seen in the macroscopic experiment (Figure 4A,B). At 0.5μg/mL of benomyl, the growth rate of the *Δgluc31* mutant strain started to decrease, although it was still higher than the parental strain ALL42. Moreover, the parental strain also presented very low growth in 1.5μg/mL of benomyl, in contrast to the *Δgluc31* strain (Figure 4C). Altogether, our results showed that *Δgluc31* mutant strain is more resistant to treatment with the fungicide benomyl, probably by the high accumulation of chitin and glucan in the cell wall of the mutant strain (Figure 2).

### 3.5. The β-Glucanase Encoded by the gluc31 Gene Is Not Required for the Potential Antagonism of T. harzianum

A gene expression analysis was conducted to verify the activity of the *gluc31* gene during the antagonism of *T. harzianum* (Figure 5A,B). The *gluc31* gene was found to be differentially expressed during the confrontation between *T. harzianum* and plant pathogens. The *gluc31* gene was found to be upregulated during the antagonism between *T. harzianum* and the phytopathogens *R. solani* and *S. sclerotiorum* under the contact condition (Figure 5C). On the other hand, under the same conditions, *gluc31* was not upregulated during contact between *T. harzianum* and *F. oxysporum*. Furthermore, *gluc31* expression remained constant during the antagonism between *T. harzianum* and *R. solani* in the post-contact conditions, whereas *gluc31* expression was downregulated during contact between *T. harzianum* and *S. sclerotiorum* under the same conditions. Moreover, *gluc31* levels were found to be downregulated during the confrontation between *T. harzianum* and *F. oxysporum* relative to those under the contact condition (Figure 5D). The efficiency in the antagonism between the wild-type (ALL42) and mutant (*Δglu31*) *T. harzianum* strains and the phytopathogens tested did not show significant differences when evaluated using the scale described by Bell et al. (1982) [30] in combination with the statistical analyses (Figure 5, Appendix A). The above results suggested that Gluc31 is not required for antagonism against plant pathogens.

To verify the involvement of Gluc31 in the compensatory response during mycoparasitism, we evaluated the enzymatic activities of glucanase, NAGase, and lichenase, as well as gene expression of chitin synthase and glucan synthase during cultivation of ALL42 and *Δgluc31* following exposure to the *R. solani* cell wall (Figure 6). After 48 h of incubation, we observed no difference in chitinase activity, but observed a significant increase in NAGase activity in the *Δgluc31* strain, as well as upregulation of chitin synthase (~40-fold) expression relative to the wild-type ALL42, thereby suggesting that the absence of Gluc31 did not influence mycoparasitism but influenced cell wall remodeling (Figure 2A–C). As expected, the total β-1,3-glucanase activity was lower in the *Δgluc31* strain than that in the ALL42 strain (Figure 6D). Furthermore, the activity of lichenase (an enzyme that catalyzes the hydrolysis of (1->4)-β-d-glucosidic linkages in β-d-glucans containing (1->3)-bonds and (1->4)-bonds) was reduced and the glucan synthase levels were upregulated (~40-fold) in the Δ*gluc31* strain compared to those in the parental ALL42 strain (Figure 6F). The results verified the compensatory regulation of cell wall remodeling in response to the absence of Gluc31 even during the mycoparasitism process.

### 3.6. Effect on GH16 Family Gene Expression in Δgluc31 T. harzianum

To analyze the phylogeny of the GH16 family in the *Trichoderma* genus, we obtained 100 genes containing the GH16 domain in the following seven species: *T. harzianum* (Triha), *T. virens* (Trivi), *T. atroviride* (Triat), *T. asperellum* (Trias), *T. reesei* (Trire), *T. citrinoviride* (Trici), and *T. longibrachiatum* (Trilo) from the MycoCosm portal [31]. The amino acid sequences of the GH16 domain were used to reconstruct their phylogeny using the maximum likelihood approach. Figure 7 shows the generated phylogenetic tree and gene domain configuration. The GH16 proteins contained 207–783 amino acids and were relatively conserved within the *Trichoderma* genus. The deleted *gluc31* gene (ID 150678) had orthologs present in all *Trichoderma* species with almost identical domain position and signaling peptide (Figure 7), which suggested that *gluc31* comprises the core gene set of *Trichoderma* and plays an important role in the genus lifestyle. These findings were confirmed by the strong negative selection observed in the alignment of the coding sequences of these seven orthologs (average pairwise nonsynonymous differences (dN) / synonymous differences (dS) dN/dS = 0.23 ± 0.06). 

To analyze the effects of the deletion of the *gluc31* gene in mycoparasitism, we performed a direct confrontation assay against the phytopathogen *R. solani* in Petri dishes. Our results showed that among these 14 genes, seven were upregulated (relative expression normalized with tested strain grown alone >2) in the WT strain during contact with *R. solani*. By contrast, GH16 genes in the Δ*gluc31* strain were downregulated during the confrontation, except for gene 84167, which was found to be upregulated relative to the WT strain. In the WT strain, genes 15000, 503986, and 513340 showed the highest expression levels under the contact condition (14-, 6-, and 15-fold, respectively). The expression levels of these genes, however, were not altered in the *Δgluc31* mutant strain, which indicated their importance during the antagonism between *T. harzianum* and *R. solani*. Interestingly, the majority of induced GH16 genes were grouped by homology in the WT (Figure 8), thereby suggesting a possible role of this subgroup in mycoparasitic interactions. Our findings suggested that the absence of gluc31 results in inhibition of the glucanolytic system.

## 4. Discussion

The *gluc31* gene, which encodes an endo-β-1,3-glucanase in *T. harzianum*, showed similarity to the EST reported in a previous study by Steindorff et al. (2012) [20], in which *T. harzianum* was grown in a culture medium containing the *F. solani* cell wall as the carbon source. Sequence prediction, however, revealed a high sequence identity of Gluc31 with enzymes from the GH16 family of all *Trichoderma* species whose genomes have been sequenced.

The roles of enzymes in cell wall formation in different yeast species have been previously described [37]. No transmembrane domain was identified for Gluc31, which suggested its location outside the membrane. In addition, the presence of thirteen potential *O*-β-glycosylation sites in Gluc31 is an indication of its activity. *O*-glycosylation is usually observed in cell wall proteins [38].

In the present study, we performed a gene deletion approach for functional analysis of the *gluc31* gene of *T. harzianum*. The replacement of the gene of interest by a marker gene via homologous recombination is considered an effective strategy for studying gene function based on phenotypic changes and has been successfully performed in various organisms, including yeast, plants, and bacteria [39].

In the present study, we reported the role of *gluc31* in cell wall formation. The morphology of the *T. harzianum* cell wall after deletion of the *gluc31* gene was evaluated via fluorescence microscopy. The mutant *gluc31 T. harzianum* cells were observed to have larger hyphae with thicker cell walls and increased deposition of chitin in apical hyphae. Chitin and glucan contents of the cell wall were evaluated in the mutant strain, and results showed that *gluc31* deletion increased chitin and glucan levels. The gene expression analysis via qRT-PCR revealed that chitin synthase and glucan synthase expression was upregulated in the mutant strain. These results demonstrated that the upregulation of chitin and glucan levels is a compensatory effect for the absence of *gluc31*. Deletion studies of the *bgl2* gene, which encodes an endo-β-1,3-glucanase from *Saccharomyces cerevisiae*, resulted in chitin accumulation in the cell wall, followed by an increase in chitin synthase activity as a repair mechanism for the damage to the cell wall [40]. High chitin levels were also observed in the *Aspergillus fumigatus* mutant for the glucanase synthase gene (*fks1*) to stabilize the cell wall and prevent lysis at the hyphae tips [41]. This compensatory effect has been well described for *S. cerevisiae* harboring mutations in genes encoding cell wall components [42,43]. Fungi have a system that ensures the integrity of the cell wall. In particular, cell wall abnormalities are detected by Wsc transmembrane sensor proteins, which are involved in cell wall integrity and stress response. The MAPK signaling cascade acts downstream of the Wsc proteins and results in the activation of a transcriptional factor that regulates the expression of genes related to cell wall biogenesis, such as chitin synthases and glucan synthases [44]. The observed characteristics of the *T. harzianum Δgluc31* strain reflect the common effect of most mutations in genes involved in cell wall organization in various fungi [45]. Functional redundancy among β-1,3-glucanases is considered responsible for the maintenance of the phenotypes of mutant lineages for *eng1* and *eng2*, which encode an β-1,3-glucanase of the GH81 and GH16 family, respectively. Sequential deletions of members of the same family are suggested to be useful in the functional study of these genes related to the cell wall of *Aspergillus fumigatus* [46].

The cell wall is a dynamic and complex structure which confers protection and rigidity to the cell and it is essential to most aspects of the fungal biology [47]. In *Candida albicans*, the resistance for drugs is associated with the upregulation of genes encoding efflux pumps. For example, high mRNA levels of the *Candida* drug resistance gene family (CDR), which is a member of the ATP-binding cassette (ABC) transporter superfamily, as well as MDR1, an MFS transporter associated with azole resistance have already been demonstrated [48]. Additionally, the mechanism of azole resistance has also been proposed to be mediated by a change in cell wall components in *C. albicans* [48]. Our results showed that deletion of *gluc31* in *T. harzianum* promoted an increase in chitin accumulation in the cell wall of the mutant strain. We also described an increase in the β-1,3-glucan content of Δ*gluc31*, pointing to a general reorganization of cell wall of the *Δgluc31* mutant strain. Many reports have shown that chitin deposition is an essential element of the restoration of cell wall structural integrity [49,50,51]. Li et al. (1999) [52] demonstrated that perturbation of chitin synthesis by the addition of nikkomycin (an inhibitor of chitin synthesis) results in an increased sensitivity to itraconazole. Together, our results suggest that the high level of chitin and glucan in the cell wall of the mutant strain might be responsible for the fungal resistance to benomyl.

A transcriptome analysis was conducted to verify the role of *gluc31* in the antagonism of *T. harzianum*. Results revealed that *gluc31* was differentially expressed during and after antagonism of the wild-type *T. harzianum* with different phytopathogens, thereby indicating the involvement of Gluc31. Similarly, β-1,3-glucanase of the GH16 family was found to be upregulated during the mycoparasitism of *T. atroviride* with *R. solani* [53]. In vitro comparisons demonstrated that the mutant (*Δgluc31*) and wild-type *T. harzianum* strain showed similar antagonistic activities. Evidence suggests that *gluc31* does not play a critical role in *T. harzianum* antagonism, which may be associated with the number of glucanase genes present in the *Trichoderma* genome [54,55]. An analysis of the various β-1,3-glucanase genes in the GH16 family demonstrated an increase in the expression of one gene in the in vitro antagonism of the Δ*gluc31* mutant strain. Our findings indicated a possible equilibrium attempt among glucanases to compensate for the lack of the *gluc31* gene, considering that these hydrolytic enzymes are important for the antagonism of *Trichoderma* spp. against phytopathogens.

## 5. Conclusions

Based on our findings on the *T. Harzianum gluc31* gene, together with those of previous studies that investigated the fungal β-1,3-glucanases, we proposed a functional gene study based on the homologous recombination knockout technique. Although present during antagonism of *T. harzianum* against various plant pathogens, *gluc31* was not found to play a crucial role in this process. In addition, the observed characteristics of the mutant strain are considered common to the deletions performed with fungal cell wall-related genes, thereby suggesting its putative function in the cell wall dynamics, which is corroborated by the results of in vitro analysis of Gluc31 function. The present study provided a differential view of the activities of β-1,3-glucanases, which are widely recognized for their roles in the mycoparasitism of *Trichoderma* spp.

## Figures and Tables

**Figure 1 biomolecules-09-00781-f001:**
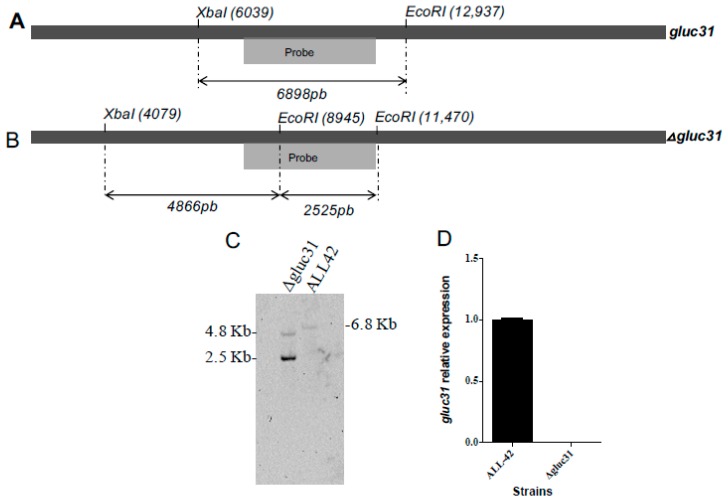
Southern Blot analysis. (**A**) Representation of *gluc31* encoding gene in *Trichoderma harzianum* wild-type strain (ALL42) and (**B**) representation of *Δgluc31* deletion cassette. *Xba*I and *Eco*RI restriction sites, and the fragments formed after digestion with their respective sizes and probe hybridization are indicated. (**C**) Southern blot analysis of total DNA from *T. harzianum* wild-type strain (ALL42) and *T*. *harzianum gluc31* mutant (*Δgluc31*) digested with *Xba*I and *Eco*RI endonuclease. (**D**) RT-qPCR from *gluc31* expression.

**Figure 2 biomolecules-09-00781-f002:**
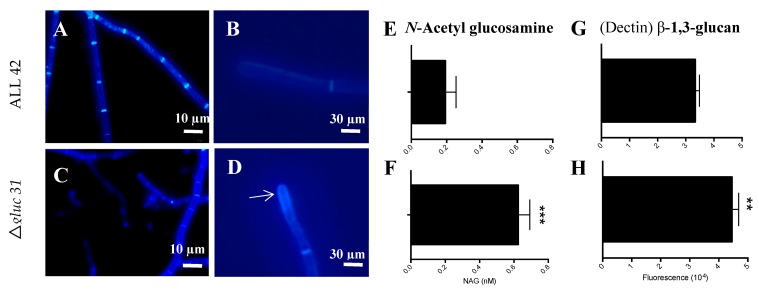
Fluorescence microscopy of *T. harzianum* ALL42 (**A**,**B**) and *Δgluc31* (**C**,**D**) stained with calcofluor white. Arrow shows chitin accumulation in apical hyphae. Chitin and glucan content of *T. harzianum* ALL42 (**E**,**G**) and *Δgluc31* (**F**,**H**). Chitin and glucan content were determined based on NAG and glucan (by dectin) detection, respectively. Values show the mean of three replicates, and the error bar indicates the standard deviation. ** *p* < 0.01, *** *p* < 0.001.

**Figure 3 biomolecules-09-00781-f003:**
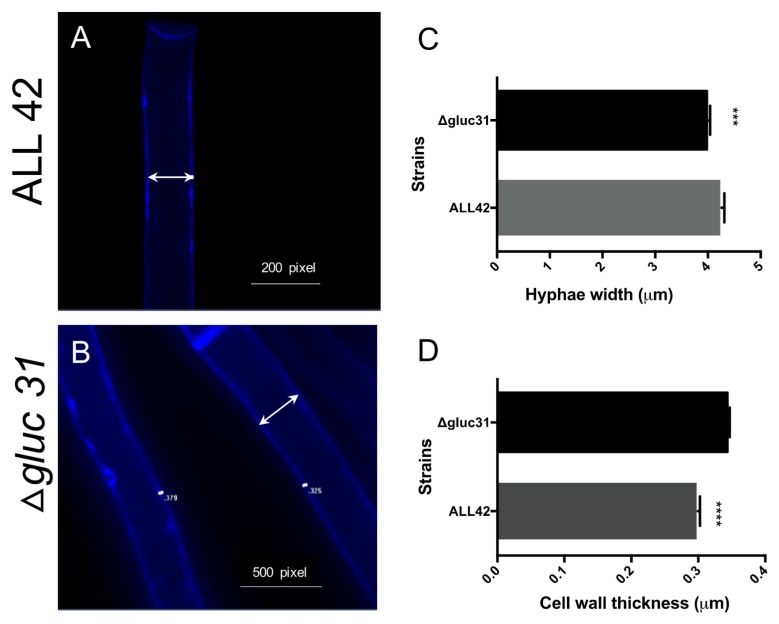
Fluorescence microscopy of *T. harzianum* ALL42 (**A**) and *Δgluc31* (**B**) stained with calcofluor white. Arrows show hyphae width. (**C**) Hyphae width of *T. harzianum* ALL42 and *Δgluc31.* (**D**) Cell wall thickness of *T. harzianum* ALL42 and *Δgluc31.* Values show the mean of three replicates, and the error bar indicates the standard deviation. *** *p* < 0.001, **** *p* < 0.0001.

**Figure 4 biomolecules-09-00781-f004:**
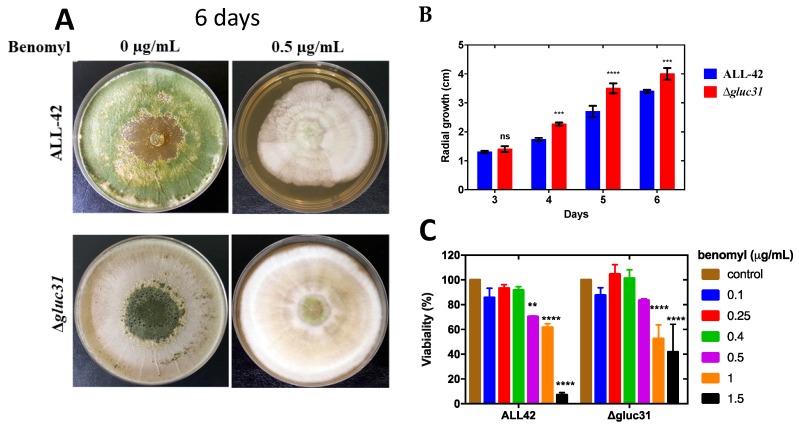
Phenotypic characterization and viability assay of parental strain (ALL42) and Δ*gluc31* under benomyl treatment. (**A**) Growth of the *T. harzianum* parent strain and Δ*gluc31* on PDA plates with 0 and 0.5 µg/mL of benomyl. The growth of the mutant and parental strains was analyzed after six days of cultivation. Data from three biological replicates. (**B**) Radial growth of the *T. harzianum* parent strain and Δ*gluc31* on PDA plates with 0.5 µg/mL of benomyl. The growth of the mutant and parental strains was analyzed after 3–6 days of cultivation. Data from three biological replicates. (**C**) Viability assay with Alamar blue for the *T. harzianum* parent strain and Δ*gluc31* treated with different benomyl concentrations. Values show the mean of three biological replicates. The error bar indicates the standard deviation (ns = non-significant, ** *p* < 0.05, *** *p* < 0.01, **** *p* < 0.001).

**Figure 5 biomolecules-09-00781-f005:**
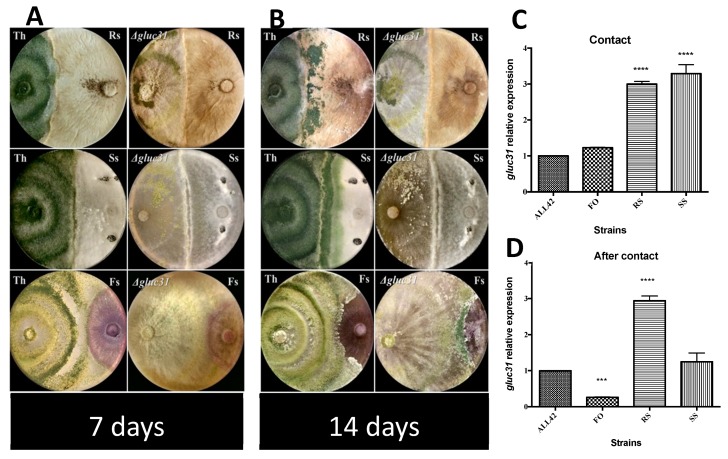
In vitro antagonistic potential assay after 7 days (**A**) and 14 days (**B**) of cultivation. (Th) *T. harzianum* ALL42; (*∆gluc31*) *T. harzianum ∆gluc31*; (Rs) *Rhizoctonia*
*solani*; (Fs) *Fusarium solani*; and (Ss) *Sclerotinia*
*sclerotiorum*. qPCR expression analyses of the *gluc31* gene during hyphae contact (**C**) and after contact (**D**). Values show the mean of three replicates. The error bar indicates the standard deviation. *** *p* < 0.01, **** *p* < 0.001.

**Figure 6 biomolecules-09-00781-f006:**
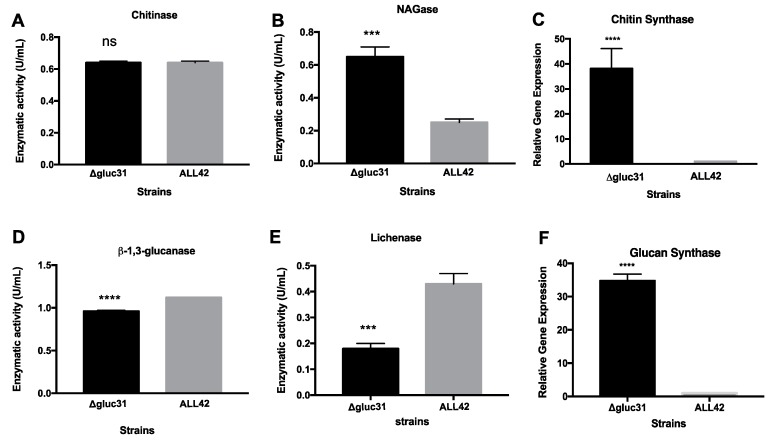
Enzymatic activity of (**A**) chitinase, (**B**) NAGAse, (**D**) β-1,3-glucanase, and (**E**) lichenase and gene expression analysis of (**C**) chitin synthase and (**F**) glucan synthase after 48 h of culture of *T*. *harzianum* ALL42 and *Δgluc31* in the presence of cell wall of *Fusarium oxysporum*. Values show the mean of three replicates. The error bar indicates the standard deviation. *** *p* < 0.01, **** *p* < 0.001.

**Figure 7 biomolecules-09-00781-f007:**
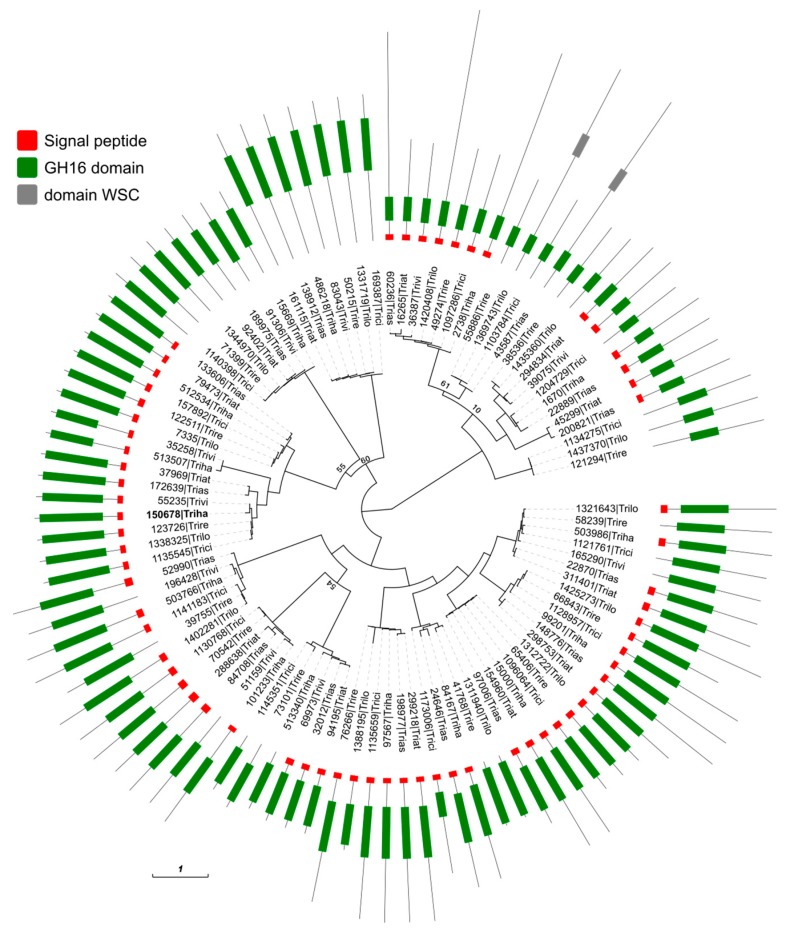
Maximum likelihood tree of the glycoside hydrolase family 16 domain (PF00722) from seven *Trichoderma* species: *T. reesei* (Trire), *T. citrinoviride* (Trici), *T. longibrachiatum* (Trilo), *T. asperellum* (Trias), *T. atroviride* (Triat), *T. virens* (Trivi) and *T. harzianum* (Triha). Only nodes with bootstrap value < 75 are shown. The red block in the gene model represents the signal peptide predicted by SignalP4.1; the green block, glycoside hydrolase family 16 domain; and the grey block, WSC domain (PF01822). The deleted gene is displayed in bold (150678).

**Figure 8 biomolecules-09-00781-f008:**
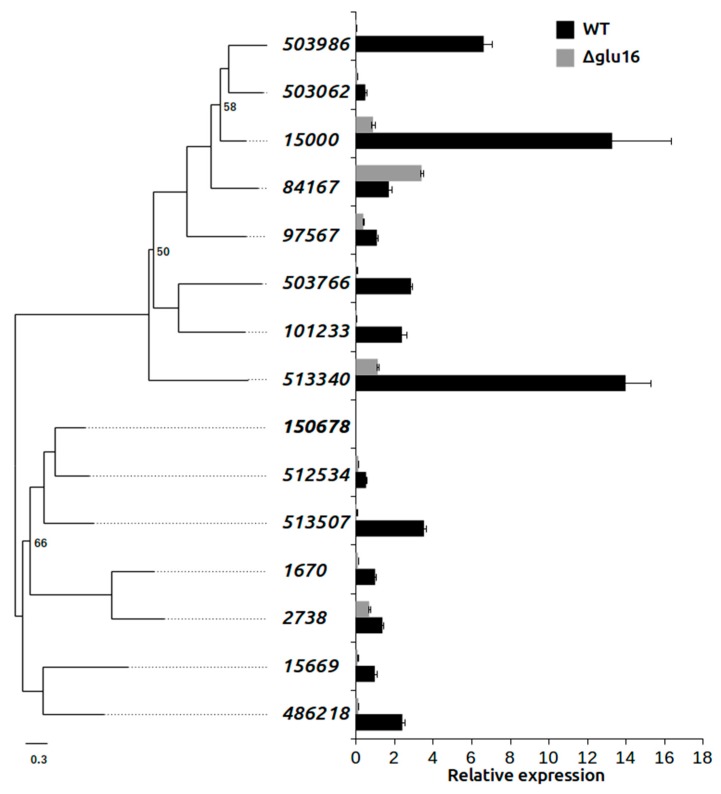
Maximum likelihood tree of the glycoside hydrolase family 16 domain from *T. harzianum*, and relative expression of the ALL42 wild-type (WT) and mutant *Δgluc31* in confrontation with *R. solani* normalized with each strain grown alone. The deleted gene is displayed in bold (150678).

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
