# Peer review of "Endo-β-1,3-glucanase (GH16 Family) from Trichoderma harzianum Participates in Cell Wall Biogenesis but Is Not Essential for Antagonism Against Plant Pathogens"

_biomolecules, 2019, doi:10.3390/biom9120781_

Round 1

Reviewer 1 Report

The manuscript by Ribeiro et al. describes a comprehensive study on an endoglucanase, gluc31, on its roles in cell wall production and mycoparasitism. The major findings were that the enzyme is involved in wall growth but is not essential for parasitism or antagonism of other fungi. The experimental approach was thorough and the manuscript is clearly presented. The following suggestions are offered for improvement.

In the Discussion, compare the present findings with previous work that some glucanases do have significant roles in antagonism of other fungi, eg. Migheli et al. 1998 Phytopathol. Complete the Reference section with page numbers for the articles cited. Typographical errors: (a) Italicize “T. harzianum” and other species throughout the manuscript. (b) add superscripts, e.g. line 187. (c) space properly, e.g. lines 241 and 244. Line 134, replace “sent” with “transferred”. Delete Figure 1E and refer to Figure 4A? The pictures show the same phenotypes of the mutant and the wild type. The legend for Figure 3 belongs to Figure 2. Move the legend to Figure 2 and add a correct one for Figure 3. Figure 4. The growth measurements shown in 4B do not agree with the observations in 4A where the colonies almost fill a 10cm plate after 6 days. Check the numbers? Figure 7. Split into two figures? Two very different observations are presented and the combination of both into one figure detracts from both messages. Figure 4 and text. The results presented in the bar graph of 4C do not support the claim that the mutant has increased resistance to benomyl. The growth of the mutant in the 0 ug/mL control is greater than the wild type as well as in the presence of the drug. However, the growth of the mutant appears to drop off at 0.5 ug/mL, before the wild type, and decreases more severely than the wild type in higher concentrations. So, at best, this characteristic is not changed. Please address.

Author Response

We really appreciated your reviewing and we address the points raised.

The manuscript by Ribeiro et al. describes a comprehensive study on an endoglucanase, gluc31, on its roles in cell wall production and mycoparasitism. The major findings were that the enzyme is involved in wall growth but is not essential for parasitism or antagonism of other fungi. The experimental approach was thorough and the manuscript is clearly presented. The following suggestions are offered for improvement.

In the Discussion, compare the present findings with previous work that some glucanases do have significant roles in antagonism of other fungi, eg. Migheli et al. 1998 Phytopathol.

AW: We appreciated your suggestion, however we would like to focus only in beta1,3 glucanases.

Complete the Reference section with page numbers for the articles cited.

AW: although we have used mendley tools, we have revised all references, thanks for your attention to that.

Typographical errors: (a) Italicize “T. harzianum” and other species throughout the manuscript. (b) add superscripts, e.g. line 187. (c) space properly, e.g. lines 241 and 244. Line 134, replace “sent” with “transferred”.

AW: corrected properly

Delete Figure 1E and refer to Figure 4A? The pictures show the same phenotypes of the mutant and the wild type.

AW: thanks for your suggestion. We deleted and referred as suggested.

The legend for Figure 3 belongs to Figure 2. Move the legend to Figure 2 and add a correct one for Figure 3. Figure 4. The growth measurements shown in 4B do not agree with the observations in 4A where the colonies almost fill a 10cm plate after 6 days. Check the numbers? Figure 7. Split into two figures? Two very different observations are presented and the combination of both into one figure detracts from both messages.

AW: we appreciated your suggestion and all of them were done. Concern to figure 4B, the figure were revised and referred as radial growth.

Figure 4 and text. The results presented in the bar graph of 4C do not support the claim that the mutant has increased resistance to benomyl. The growth of the mutant in the 0 ug/mL control is greater than the wild type as well as in the presence of the drug. However, the growth of the mutant appears to drop off at 0.5 ug/mL, before the wild type, and decreases more severely than the wild type in higher concentrations. So, at best, this characteristic is not changed. Please address. 

AW: We really appreciated your opinion and indeed is right. We performed the experiment again, initiating the measurements in 3 days of growth. In relation of Figure 4C we expressed the results as relative (%) of viability to exclude the interference of different rates of growing in the strains. Now is more reliable.

English minor spell was corrected by professional company.

Reviewer 2 Report

The manuscript entitled "Endo-β-1,3-glucanase (GH16 family) from
Trichoderma harzianum participates in self-cell wall biogenesis but is not essential for antagonism against plant pathogens" reports on the role of the gluc31 gene, which encodes an endo β-1,3-glucanase belonging to the GH16 family in Trichoderma harzianum. The authors found that the
absence of the gluc31 gene did not affect the in vivo mycoparasitism ability
of the mutant T. harzianum. However, gluc31 influenced cell
wall organization. Moreover, the mutant strain became more resistant to the fungicide Benomyl compared to the wt. I find the work interesting from the point of view of the functional genomics of plant-friendly fungi. The experimental setup is correct, and the experimental level presented is suitable for this journal. The work is well written apart from some errors in the presentation of English. For this reason, I advise the intervention of a native English speaker. I suggest implementing the introduction in the discussion of how Trichoderma harzianum helps plants strengthen their defenses by triggering a pre-immunization in the host plant.

However, I find several problems in presenting the results especially in the legends of the figures.
Revisions:
1- The description of figures 1F and H is missing in the legend. In the same figure it is not clear how the data is expressed. Repeat the field names also on the histograms.
  2- There are even 4 asterisks in figure 2D while in the legend of figure 2 there are two asterisks. To review.
3- In figure 3, there are words in probably Portuguese language.

Author Response

Dear Reviewer,

We appreciated your comments. Some English spell errors were revised by professional company and native English speaker.

Related to introduction and discussion of how Trichoderma helps plants in defense and immunization, we appreciated your suggestion, but we understand that the focus of the work is in mycoparasitism and we preferred not to include this topic at moment.

All figures legends were revised and correct properly as suggested.